# The Therapeutic Potential of Imidazole or Quinone-Based Compounds as Radiosensitisers in Combination with Radiotherapy for the Treatment of Head and Neck Squamous Cell Carcinoma

**DOI:** 10.3390/cancers14194694

**Published:** 2022-09-27

**Authors:** Abul Azad, Anthony Kong

**Affiliations:** Comprehensive Cancer Centre, School of Cancer and Pharmaceutical Sciences, King’s College London, Guy’s Campus, New Hunt House, London SE1 1UL, UK

**Keywords:** imidazole, quinone, radiosensitisers, atovaquone, head and neck squamous cell carcinoma

## Abstract

**Simple Summary:**

Patients with curable head and neck cancers are usually treated with a combination of chemotherapy and radiotherapy, but they experience significant, severe side effects, which greatly affect their quality of life. Some of these patients still experience disease relapse after an intensive course of treatment due to tumours that are resistant to radiotherapy and chemotherapy because of hypoxia (lack of oxygen). In addition, some patients are not suitable for and/or are not able to have combined chemotherapy with radiotherapy due to their age or other physical conditions. Certain small-molecule drugs, which are used to treat various infections including malaria, have been shown to reduce hypoxia and thus make radiotherapy more effective. Therefore, their combination with radiotherapy could have less toxicities compared with the combination of chemotherapy with radiotherapy. Here, we discuss the promising results from preclinical work and clinical trials of these agents, and their potential use in the clinic, to reduce hypoxia and to sensitise radiotherapy. These agents could potentially be used for patients who are not suitable for combined chemotherapy and radiotherapy; they may also be used to reduce the dose of radiotherapy if able to enhance radiotherapy effect at lower dose in order to reduce toxicities while maintaining the treatment efficacy in a more personalised manner.

**Abstract:**

The addition of platinum chemotherapy to primary radiotherapy (chemoradiation) improves survival outcomes for patients with head and neck squamous cell carcinoma (HNSCC), but it carries a high incidence of acute and long-term treatment-related complications, resulting in a poor quality of life. In addition, patients with significant co-morbidities, or older patients, cannot tolerate or do not benefit from concurrent chemoradiation. These patients are often treated with radiotherapy alone resulting in poor locoregional control and worse survival outcomes. Thus, there is an urgent need to assess other less toxic treatment modalities, which could become an alternative to chemoradiation in HNSCC. Currently, there are several promising anti-cancer drugs available, but there has been very limited success so far in replacing concurrent chemoradiation due to their low efficacy or increased toxicities. However, there is new hope that a treatment strategy that incorporates agents that act as radiosensitisers to improve the efficacy of conventional radiotherapy could be an alternative to more toxic chemotherapeutic agents. Recently, imidazole-based or quinone-based anti-malarial compounds have drawn considerable attention as potential radiosensitisers in several cancers. Here, we will discuss the possibility of using these compounds as radiosensitisers, which could be assessed as safe and effective alternatives to chemotherapy, particularly for patients with HNSCC that are not suitable for concurrent chemotherapy due to their age or co-morbidities or in metastatic settings. In addition, these agents could also be tested to assess their efficacy in combination with immunotherapy in recurrent and metastatic settings or in combination with radiotherapy and immunotherapy in curative settings.

## 1. Introduction

Head and neck cancers are heterogeneous, of which the most common histology type is squamous cell carcinoma, and they are collectively known as head and neck squamous cell carcinoma (HNSCC) [1]. Most of the HNSCC are associated with excess smoking and alcohol intake; however, human papillomavirus (HPV) infection has accounted for a rising incidence of oropharyngeal HNSCC [2]. More details about the aetiology of the disease are reviewed elsewhere [3]. Briefly, around 90% of HNSCC arise in the oral cavity, oropharynx, hypopharynx and larynx. HPV-negative HNSCC can occur anywhere in the head and neck areas with most of them being found in the larynx and oral cavity [4], whereas HPV-positive HNSCC usually occur within the oropharynx [5]. HPV-positive HNSCC respond better to radiation and/or chemotherapy and have a better outcome following treatments compared to HPV-negative diseases [6].

Although HPV-positive cases are significantly different from HPV-negative cases at the molecular level, most locally advanced HNSCC are usually treated with a combination of surgery, radiation and chemotherapy—with the tumour subsites and stage of the diseases determining the primary treatment modality [1,7]. For example, oropharyngeal cancers (regardless of HPV status) tend to be treated with primary radiotherapy (with or without concurrent chemotherapy), whereas oral cavity cancers are usually treated with surgery followed by postoperative radiation (with or without concurrent chemotherapy), depending on the risk factors. Single-modality treatment is recommended for early stage disease but a combination treatment approach is required for locally advanced disease [1]. Radiotherapy is the primary choice of treatment (beside surgery) for early stage disease, especially if organ preservation is desired, such as those arising from the larynx. Platinum chemotherapy, either cisplatin or carboplatin, is the standard type of agent used in combination with adjuvant or primary radiotherapy since its concurrent treatment with radiotherapy has been shown to improve overall survival in HNSCC; however, the survival benefit decreases with increasing age and no benefit was found in patients over 70 [8]. Targeted therapies have been developed to use alongside chemo-radiotherapy and/or replacing chemotherapy with the hope of further improving treatment efficacy and reducing toxicity. 

Overexpression of EGFR (epidermal growth factor receptor), a transmembrane protein belonging to the EGFR tyrosine kinase family, occurs in 90% of HNSCC; further, cetuximab, which binds to the extracellular region of EGFR, is the only targeted therapy approved for the treatment of HNSCC [9]. In addition to concurrent platinum chemotherapy with radiotherapy, adding cetuximab to primary radiotherapy has also been shown to increase survival compared to radiotherapy alone [10]. However, no survival outcome benefit was demonstrated when cetuximab was added to chemoradiation compared to chemoradiation alone [6,11]. In addition, two recent randomised trials (the De-ESCALaTE and TROG 12.01 studies), comparing concurrent cisplatin with cetuximab—in combination with radiotherapy in HPV-positive oropharyngeal cancer patients—have shown an inferior overall survival when cetuximab is combined with radiotherapy compared to concurrent platinum chemotherapy with radiotherapy [12,13].

Cetuximab, as a single agent, showed a response rate of less than 15% in the treatment of recurrent or metastatic HNSCC [14]. The combination of cetuximab with cisplatin chemotherapy compared to cisplatin chemotherapy in these patients resulted in an increase in the objective response rate (ORR) from 10% to 26%, but no improvement in progression-free survival (PFS) and overall survival (OS) [15]. However, the addition of cetuximab to cisplatin and the 5FU chemotherapy (EXTREME) regimen improved ORR, PFS and OS compared to cisplatin and 5FU chemotherapy alone. This resulted in its approval (previously) as the first-line treatment for recurrent or metastatic HNSCC. Cetuximab is the only approved EGFR monoclonal antibody (mAb), either in combination with radiotherapy for HNSCC patients not suitable for concurrent platinum chemotherapy, or in combination with platinum-based chemotherapy in recurrent or metastatic HNSCC, in the UK. Other EGFR mAbs, such as zalutumumab, panitumumab, nimotuzumab, have been tested in numerous clinical trials and in different settings; however, none of them have been approved for use in HNSCC in the UK [16]. 

In addition to cetuximab, anti-PD1 antibodies pembrolizumab and nivolumab are approved for treatments in recurrent or metastatic HNSCC. The KEYNOTE-048 study—a randomised phase 3 trial comparing pembrolizumab as a monotherapy or in combination with platinum chemotherapy plus 5-FU vs. to cetuximab with platinum chemotherapy plus 5-FU (EXTREME regimen)—showed that pembrolizumab monotherapy significantly improved overall survival (OS) over the EXTREME regimen in the PD-L1 combined pathology score (CPS) ≥20 and ≥1 populations. Moreover, pembrolizumab plus chemotherapy significantly improved OS in the total population, with safety comparable to the EXTREME regimen as a first-line treatment, resulting in the approval of pembrolizumab for recurrent or metastatic HNSCC as a first-line treatment as well [17]. Nivolumab has also been shown to improve OS for recurrent or metastatic HNSCC after previous platinum-based chemotherapy was compared to the investigators’ choice of second-line treatments (docetaxel, methotrexate or cetuximab [18]), resulting in nivolumab’s approval as a second-line treatment after platinum-based chemotherapy. 

A significant number of HNSCC patients, who are currently treated by radiotherapy, are over 70 years old. Although these patients were not excluded from the Bonner trial [10], many oncologists would not offer concurrent cetuximab with radiotherapy for older patients due to a concern that it would increase skin toxicities [19]. In addition, the subsequent subgroup analyses from the study showed that the benefit of cetuximab is confined to patients who are less than 70 years old [10]. Most of the older patients would receive radiotherapy alone since there were no survival benefits shown for concurrent platinum chemotherapy with primary radiotherapy in patients over 70 years old [8]. Therefore, there is a need to develop other strategies that can help to improve survival for these locally advanced elderly HNSCC patients who would otherwise receive radiotherapy alone due to their age or co-morbidities if platinum-based chemotherapy is deemed to be inappropriate. 

In this review, we highlight the most recent advances in the use of either imidazole or quinone-based compounds as radiosensitisers or as anti-cancer therapies. In particular, we wish to explore the possibility of using atovaquone (which is an FDA-approved anti-malarial drug) as a radiosensitiser, or as an alternative to concurrent platinum chemotherapy in combination with radiotherapy. We limit our discussion to treatment strategies for HPV-negative HNSCC, although this review may also be useful for HPV-positive HNSCC. 

## 2. Imidazole-Based Compounds as Radiosensitisers in the Treatment of HNSCC

Some solid tumours can be relatively refractory to radiotherapy or other treatments. One of the major reasons for treatment failure is related to new growth of the tumour at the primary site via more resistant cancer cells or clones that have survived the lethal action of radiation or other treatment and have therefore developed resistance [3]. Moreover, solid tumours frequently maintain a state of hypoxia, which would renders them more resistant to radiation damage than those in a normal oxygen environment [20,21]. Radiation is used as a primary and adjuvant (+/− concurrent chemotherapy) or palliative treatment for head and neck cancers [1], but patients often develop treatment resistance due to hypoxic tumours, resulting in a poor outcome. This provides a challenge and an opportunity to overcome this treatment resistance through an alternative measure to platinum chemotherapy since a chemical compound that reduces the hypoxic status of the tumour may enhance radiation response and could also be used as a radiosensitiser [22,23]. 

Radiosensitisers, or radio-enhancers, are chemical compounds that enhance the radiation effect of killing or inhibiting the growth of cancer cells when combined with radiotherapy. Some of the established drugs, such as platinum chemotherapy or cetuximab, have been used in combination with radiation to treat multiple types of cancers, including HNSCC, as discussed above. It is also worthwhile to exploit the concept of small-molecule chemical radiosensitisers in the treatment of head and neck carcinoma [22,23]. Recently, several compounds, such as imidazole or quinone-based compounds, have been tested as radiosensitisers in clinics in an attempt to overcome radioresistance and to enhance the radiation effect of multiple cancer treatments. 

Imidazole is a nitrogen-containing heterocyclic ring that possesses many important biological and pharmacological activities. Due to its versatile medicinal properties, imidazole has drawn considerable interest in recent years with a specific focus on its role as an anticancer therapy. Since imidazole-based molecules could halt cell division through covalent or non-covalent interactions with DNA (due to their high binding capacity to protein molecules), they are seen as an attractive therapeutic agent to control cancers [20,21]. These molecules are of great interest as anti-cancer agents due to their potential in increasing the efficacy of treatments and in producing better toxicity profiles compared to conventional chemotherapy. In addition, it is also potentially possible to lower the standard doses of radiotherapy or chemotherapy if imidazole-based molecules can be used to enhance the combination of radiotherapy and chemotherapy effects, thus rendering less toxicity to the patients.

A whole range of natural and synthetic compounds contain imidazole rings, such as dacarbazine and zoledronic acid, and are used as anti-cancer agents in patients. Details about their structure, functions and their relation to particular cancer treatments have been described previously [22]. There is another derivative of imidazole known as 5-nitroimidazole, i.e., a nitro group at position 5 of the imidazole ring, and drugs of this variety are metronidazole, tinidazole, nimorazole, dimetridazole, ornidazole, megazol and azanidazole [22,23]. Further, drugs based on 2-nitroimidazole include misonidazole, etanidazole and benznidazole [24]. Nitroimidazoles were originally developed as antibiotics to combat bacterial and parasitic infection [25]. Among these compounds, misonidazole, etanidazole, metronidazole and nimorazole have been studied extensively in preclinical and clinical studies [26,27]. A panel of compounds with similar functions can be found in [23] and clinical studies of these compounds conducted in patients with HNSCC and other cancers are listed in Table 1 and Table 2 Most of the nitroimidazole group of drugs have a wide spectrum of activity against protozoa, both Gram-negative and positive bacteria, as well as hypoxic tumours, with a low incidence of resistance despite being used in the clinic for the last 30 years. Details about their mechanism of action against anaerobes have been described previously [28]. Briefly, nitro radical anions can be generated through the reduction of nitro groups in nitroimidazoles after they enter into the cell. Such radicals can oxidise DNA via electron transfer, which induces DNA strand breaks and subsequent cell death. Although they are acting as an oxygen mimetic, the detailed mechanism on how this causes a reduction in hypoxia is not clearly known yet. In this review, we aim to update the data on nitroimidazole-based oxygen mimetics (misonidazole, etanidazole, metronidazole and nimorazole) in preclinical and clinical studies of head and neck cancer treatment. We explore the possibility of using these drugs as potential radiosensitisers in the treatment of HNSCC through reducing tumour hypoxia and overcoming radioresistance. 

### 2.1. Metronidazole as a Radiosensitiser in HNSCC Cancer

Although there are number of agents in this category, only metronidazole and misonidazole (or 2-nitroimidazole) have been investigated to a large degree in HNSCC. Metronidazole is a member of the imidazole group, and is a nitroimidazole-derivative bactericidal agent widely used in the treatment of many anaerobic and certain protozoan and parasitic infections. However, three decades ago, metronidazole was tested as a radiosensitiser with interstitial irradiation in recurrent or persistent oral cavity cancers following previous primary treatment with surgery or radiotherapy or their combination [29]. Out of 23 patients in the trial, 16 patients (69%) showed a complete regression of the local disease and there was no evidence of further recurrence in 10 patients (43%) after an average follow up of 25 months following completion of treatment. The combination treatment was well tolerated, with nausea, mild diarrhoea and mucositis being the main side effects [29]. The study was not randomised against interstitial radiation alone and therefore it is uncertain whether the clinical efficacy was due to radiation alone. The radiosensitising effect of metronidazole is dose-dependent; however, higher doses and prolonged use of metronidazole can lead to rare neurotoxicity, including encephalopathy. One study combined the use of metronidazole with local hyperthermia, which allows for higher drug concentrations in tumour tissue [39]. In this study, 64 patients with locally advanced, fixed T4 rectal cancer underwent a course of neoadjuvant, hypofractionated radiotherapy with concurrent capecitabine and oxaliplatin chemotherapy as well as local hyperthermia and metronidazole [39]. Impressively, 59 patients underwent R0 resection of the tumour and only five patients remained inoperable. The grade 3 toxicities include diarrhoea (n = 10), vomiting (n = 2), proctitis (n = 2) and skin reaction (n = 1); additionally, only one patient had grade IV diarrhoea and vomiting but no grade 3–4 neurotoxicity was observed [39]. Despite initial promising findings many decades ago, further clinical studies with metronidazole in HNSCC have not been conducted. Previous studies did not assess the hypoxia status of the tumour following the administration of metronidazole. Moreover, to date, there are no preclinical studies with this drug in mouse models of head and neck cancers. Based on the limited clinical studies available, we cannot rule out metronidazole as a useful radiosensitiser and further detailed evaluation of metronidazole as a radiosensitiser, along with its mechanism, may be required in HNSCC, especially in combination with current intensity-modulated radiotherapy (IMRT).

### 2.2. Use of Misonidazole and Etanidazole as a Radiosensitiser in HNSCC

It has been previously shown that 2-nitroimidazoles (misonidazole, etanidazole) act as radiosensitisers of hypoxic tumour cells [24]. In a European randomised trial of etanidazole with 374 patients, the addition of etanidazole (at a dose of 2 g/m^2^ for a total of 17 doses, three times per week) to conventional radiotherapy (between 66 Gy in 33 fractions and 74 Gy in 37 fractions, at 5 fractions/week) did not improve 2-year loco-regional control rates (LCR) and 2-year overall survival (OS) rates for patients with head and neck carcinoma compared to conventional radiotherapy alone [30]. There were 52 cases of grade 1–3 peripheral neuropathy in the etanidazole group (n = 187) compared to 5 cases of grade 1 peripheral neuropathy in the control group (n = 187) [30]. In another independent phase III trial (RTOG 85-27) of 521 patients, the addition of etanidazole (2 g/m^2^ for a total of 17 doses) to conventional radiotherapy (between 66 Gy in 33 fractions and 74 Gy in 37 fractions) in locally advanced HNSCC has shown no global benefit in 2-year LCR and ORR compared to conventional radiotherapy alone. However, the multivariate analysis suggested a benefit for patients who had N0-1 disease [31]. In total, 18% of patients developed grade 1 and 5% of patients developed grade 2 peripheral neuropathy in the etanidazole arm [31].

Similarly, another compound misonidazole was also tested in patients with HNSCC. In an EORTC randomised trial of 523 patients with locally advanced head and neck cancer, the addition of misonidazole (1 g/m^2^ on every irradiation day, total 12 to 14 g/m^2^) to multiple fractions per day (MFD) of radiotherapy (3 × 1.6 Gy/day for 10 days, 3 weeks rest, followed by a boost to 67.2 or 72 Gy) showed no differences in survival or locoregional control compared to MFD alone or classical radiotherapy (70 Gy/7 weeks) [32]. In another randomised trial of a DAHANCA 2 study in patients with invasive larynx and pharynx, the addition of misonidazole 11 g/m^2^ to radiotherapy did not have a significantly better LCR compared to the placebo groups; however, a better control rate was found in patients with pharynx carcinomas [33]. Moreover, in a phase I/II randomised study (RTOG 79-04) of 42 patients, with unresectable HNSCC, the addition of misonidazole (1.5 g/m^2^) with a high fractional dose of radiotherapy (4 Gy/day, 5 days per week for a total of 44–52 Gy) did not improve the efficacy of high fractional dose of radiotherapy [34]. In fact, most of the clinical trials conducted with 2-nitroimidazole turned out to be negative and these compounds have shown some concerning toxicities, such as neurotoxicity. Overall, neither etanidazole nor misonidazole had any significant effect in LCR and OS when combined with radiation and compared to radiation alone and thus may not be a safe and effective radiosensitiser to enhance the radiation response in HNSCC. One reason for this could be that the drug concentration in plasma or the tumour may not reach the level of the administration dose or threshold level to be effective as a radiosensitiser. In addition, we could not find any preclinical study assessing these compounds in combination with radiation in HNSCC mouse models. Other compounds such as tinidazole, dimetridazole, ornidazole, megazol, benznidazole and azanidazole do not seem to have been evaluated as hypoxic radiosensitisers in either preclinical or clinical settings. 

### 2.3. Possibility of the Use of Nimorazole as a Radiosensitiser in HNSCC

As stated above, the use of the 2-nitroimidazole hypoxic radiosensitiser misonidazole showed no, or limited, clinical benefit and its neurotoxicity severely restricted the total dose when combined with radiotherapy [34,40]. Compared to misonidazole, patients treated with 5-nitroimidazoles did not develop serious or long-lasting side effects, especially in regard to neurotoxicity [41]. The only member in this group of compounds that is currently approved for clinical use is nimorazole (5-nitroimidazole), which is used for subgroups of patients with advanced head and neck cancers in Europe. In a randomised trial of a phase III study (DAHANCA 5) of 422 patients, nimorazole in addition to radiotherapy (62–68 Gy, 2 Gy per fraction, 5 fractions/week) showed a significant improvement in LCR (49% vs. 33%) and a non-significant improvement in OS (when compared to a placebo in patients with pharynx and supraglottic larynx tumours) [35]. In a phase II study of 61 patients with stage III and IV head and neck cancers, the patients were treated with combined continuous hyper-fractionated accelerated radiotherapy (CHART) and nimorazole, of which the overall 2-year LCR was 55%. This is numerically higher than those previously seen with CHART, suggesting a potentially addictive effect of nimorazole [36]. In another randomised multicentre trial (IAEA-HypoX), the addition of nimorazole (at a dose of 1.2 g/m^2^ administered over 90 min) before the first daily radiation fraction (66–70 Gy in 33–35 fractions and 6 fractions/week) seemed to improve numerical LCR and decreased the death rate. However, this was not statistically significant due to the trial being terminated prematurely as a result of poor recruitment [42]. 

In a more recent study that evaluated the effect of hyperfractionated accelerated radiotherapy (HART) at a dose of 76 Gy in 56 fractions (10 fractions weekly) with nimorazole in 295 patients with unresected HNSCC (larynx, pharynx and oral cavity), the 3-year LRF of 19% and OS of 66% were comparable to treatment with acceleration and/or chemo-radiation of the historical data from previous DAHANCA trials [37]. In the biomarker study of a DAHANCA 5 trial using p16-status as a retrospective stratification, it was shown that nimorazole in addition to radiotherapy improved loco-regional control in HPV p16 negative patients but not HPV p16-positive patients in the subgroup analysis. This suggests that hypoxic radioresistance may not be as clinically relevant in HPV-positive tumours [43]. However, from the results of the above trials, it has been demonstrated that nimorazole can be administered without major side effects or toxicity. 

Apart from the HART trial [37], all of the above trials were conducted many years ago and there were not many randomised trials comparing the combination of radiotherapy and nimorazole against radiotherapy alone. A multicentre study was conducted in the UK, called the NIMRAD study, which is a randomised placebo-controlled trial of synchronous nimorazole with radiotherapy versus radiotherapy alone in patients with locally advanced HNSCC that are not suitable for concurrent chemotherapy or cetuximab treatment. This study has completed recruitment and the results are eagerly awaited. In short, several clinical trials that investigated the addition of misonidazole or etanidazole to radiotherapy failed to show benefits in LCR or OS in patients, while the addition of nimorazole to radiotherapy appeared to have some benefits. Despite the potential clinical benefits of nimorazole as a radiosensitiser in head and neck cancers, it is not routinely used in clinical practice worldwide, except in Denmark [44].

In in vitro and in vivo studies with a xenograft model of oesophageal adenocarcinoma, it was shown that nimorazole enhanced the efficacy of radiation in suppressing cancer cell proliferation in hypoxic tumour areas but it did not seem to improve tumour growth control [45]. In another study, nimorazole sensitised radiotherapy in hypoxic conditions in squamous cell carcinoma (SSCVII in C3H mice) xenograft model but not in aerobic conditions. In this study, they also compared nimorazole with two more compounds—sanazole (3-nitrotriazole) and KU-2285 (fluorinated 2-nitroimidazole derivative)—for their radiosensitising capacities. Although KU-2285 has a superior capacity in radiosensitisation compared to both nimorazole and sanazole, KU-2285 may not be utilised as a radiosensitiser due to the difficulty and high cost involved in its synthesis. There were no significant differences between sanazole and nimorazole; however, sanazole appeared to be slightly more efficient than nimorazole in tumour growth delay [46]. Sanazole is distributed by the International Atomic Energy Agency, whereas nimorazole is available commercially. 

Moreover, Overgaard et al. (1982) investigated nimorazole in comparison with misonidazole and found that nimorazole was almost as efficient in radiotherapy as misonidazole at a lower drug dose of 100 mg/kg in murine tumours but less efficient than misonidazole at higher doses [27]. There were some discrepancies between the results obtained in clinical studies when compared against preclinical studies. They found that misonidazole was superior to nimorazole in an in vivo murine model but there were no clinical benefits demonstrated with the addition of misonidazole to radiotherapy for patients in clinical trials. However, nimorazole was demonstrated to have clinical benefits as a radiosensitiser despite limited success in combination with a high radiation dose in mouse xenograft models. In fact, most of the above-mentioned studies have not offered mechanistic insights into the radiosensitising effect. Dose scheduling is likely to be very important in determining the treatment outcome, even though the radiation dose was similar in all the above-described nimorazole clinical trials. The promising clinical findings with DAHANCA studies have somehow been lost in a crowd of negative findings with other nitroimidazole compounds, perhaps because most of these were older-generation compounds. Further well-conducted studies are required to rule out these discrepancies (specifically with different dose scheduling) in both preclinical and clinical settings. It will also be important to investigate the mechanism of reduction in hypoxia, which has not been clearly demonstrated previously. Moreover, any oncogenic signalling pathways that are affected by nimorazole are yet to be uncovered. 

As discussed, misonidazole is unsuitable for further clinical use due to the development of neuropathy in patients and because 5-nitroimidazoles, such as metronidazole, are known to be less efficient than 2-nitroimidazoles. Additionally, the effects of 3-nitroriazole sanazole have not been well studied compared to 2-nitoimidazoles. Considering the relatively potent in vivo effects and the minimal toxicity of nimorazole, it seems reasonable to plan further studies with nimorazole as a radiosensitising agent in clinics.

Nitroimidazoles have been extensively explored as hypoxic cell radiosensitisers but have had limited clinical success as their efficacy is restricted by their toxicities. The four nitroimidazole compounds discussed in this review have different nitroheterocyclic rings, but their mechanism of action (i.e., electron-affinic sensitisation) is considered to be similar [47]. 

## 3. Quinone-Based Compound in the Treatment of HNSCC: Atovaquone 

Atovaquone is a chemical compound that belongs to the class of naphthoquinone analogues of ubiquinone. In the US, it is available as an oral suspension, under the brand name Mepron, and it is also available as an atovaquone suspension, which is an FDA-approved drug used for the treatment and prevention of malaria [48]. Details about its structure and other physical chemistry features have been described previously [49]. Briefly, the structure of atovaquone is depicted in Figure 1 and it is thought to act as a potent and selective OXPHOS (oxidative phosphorylation) inhibitor, by targeting mitochondrial complex III (Figure 2). It is a highly lipophilic compound with a very limited aqueous solubility; moreover, it is protein bound (>99%), but causes no significant displacement of other highly protein-bound drugs. Atovaquone is structurally similar to the inner mitochondrial protein coenzyme Q (also known as ubiquinone), which is an integral component of the electron transport chain in oxidative phosphorylation. Coenzyme Q accepts electrons from dehydrogenase enzymes and passes them to cytochrome *bc*_1_ (complex III), which requires the binding of coenzyme Q-complex III at the Qo cytochrome domain. However, atovaquone inhibits this step of electron transport chain [49,50,51]. The consequence of this inhibition is the collapse of mitochondrial membrane potential and thus the rendering of apoptosis [52]. 

In a recent study, Ashton et al. (2016) demonstrated that atovaquone could alleviate tumour hypoxia at pharmacologically achievable concentrations by reducing oxygen consumption rate (OCR) [53]. By slowing down the use of oxygen, atovaquone reversed low oxygen levels in tumours. As a consequence, they found that atovaquone enhanced radiation response to FaDu and HCT116 xenograft tumour models as well as in multiple cell line spheroids [53]. In another study, it has been shown that atovaquone has anti-tumour activity in a multiple myeloma xenograft, which is due to the inhibition of STAT3 [54]. In a separate preclinical study by Stevens et al. (2019), it was shown that atovaquone induced apoptosis with acute myeloid leukaemia (AML) cell lines and primary pediatric AML specimens through the inhibition of mTOR activity [55]. They also demonstrated that atovaquone decreased disease burden and prolonged survival in NSG mice, which were xenografted with luciferase-expressing THP-1 cells, and in those that received a patient-derived xenograft [55]. Atovaquone was also shown to significantly reduce the expression of HER2 and its downstream effector in a panel of breast cancer cell lines and mouse models of C166 and 4T1 breast cancer cells, resulting in a reduction in tumour growth in these models [56]. In addition, atovaquone showed anti-cancer activity in cancer stem cells [57]. 

Moreover, it has been shown that atovaquone can also induce apoptosis in hepatocellular carcinoma (HCC) by a mechanism of DNA double-stranded breaks both in vitro and in vivo [58]. They have shown that atovaquone inhibited cell proliferation via S phase cell arrest and induced apoptosis through the upregulation of p53 and p21 [58]. Additionally, atovaquone was also effective against cervical cancer cells via the inhibition of complex III of mitochondrial respiration [59]. The possibility of using several other OXPHOS inhibitors in cancer treatment has been reviewed in detail [60]. In this review, we do not cover OXPHOS inhibitors in detail; rather, we explore some of their uses, in particular atovaquone, as potential radiosensitisers or as alternatives through which to target cancer cells therapeutically. From the aforementioned in vitro and in vivo studies, it is clear that atovaquone may have multiple targets ranging from oncogenic signalling and the cell cycle, as well as being an OXPHOS inhibitor specific to different cancers. Thus, it warrants further detailed preclinical and clinical studies to demonstrate its radiosensitising capabilities with detailed mechanisms.

In addition, it is worth mentioning that metformin, an anti-diabetic drug, though not a quinone-based compound, acts as an OXPHOS inhibitor through complex I inhibition (Figure 2) [61]. It was shown to alleviate hypoxia in vivo through reducing OCR and results in the inhibition of the mTOR1 pathway through AMPK activation [61]. It was shown that metformin enhanced the radiation response, which reduced tumour growth in several cancer models [53,62]. Due to these results, metformin has been tested in several clinical trials. 

It is a very remarkable finding that an anti-malarial drug could be repurposed in the treatment of cancer as a radiosensitiser. The development of safe new anti-tumour agents has become increasingly important due to the steady rise in drug-resistant tumours. In a recent small-window-of-opportunity study of atovaquone (NCT02628080) as a tumour hypoxia modifier, it was demonstrated that atovaquone could reduce tumour hypoxia [38] in patients who were referred for surgery with suspected non-small cell lung cancer (NSCLC). As atovaquone can reduce hypoxia, this enhances the radiation response to tumour growth in a xenograft model [53]. A phase 1 safety study (Arcardian) is now ongoing to investigate whether atovaquone can be safely combined with chemoradiation in patients with NSCLC. Apart from atovaquone, there are other quinone-based compounds that have also been tested in clinical trials and reviewed in detail, albeit with no significant outcomes [63].

## 4. Conclusions

The negative impact of hypoxia on treatment efficacy acts as a significant barrier to improve clinical outcomes in patients. Historically, numerous approaches to alleviate tumour hypoxia have been explored; however, none are in widespread clinical use today. Most strategies were predicated on therapeutic agents reaching hypoxic areas that are difficult to penetrate, yielding minimal or no clinical efficacy. Attempting to increase oxygen supply directly via hyperbaric chambers was similarly ineffective. The use of arsenic acid was found to be associated with significant toxicity, although it was shown to reduce hypoxia in animal models. In this review, we highlighted the potential use of nimorazole, atovaquone or metformin as hypoxia modifiers in clinical settings since these drugs can be used without significant toxicities, while also being able to sensitise radiotherapy. It must be said that there is still limited information about the use of atovaquone in reducing tumour hypoxia in clinical settings, especially in HNSCC. Nimorazole is recommended in HNSCC in Denmark only. Future studies with these drugs in both preclinical and clinical settings will help to determine whether they would be useful in alleviating hypoxia and enhancing radiotherapy or the chemo-radiation response. In addition to the radiosensitising effect, it may also be useful to consider these agents in other combinations such as in immunotherapy, especially in patients with recurrent or metastatic HNSCC. In addition, if there are further promising data, it may be possible to investigate other uses of these hypoxia modifiers in combination with radiotherapy and in locally advanced HNSCC in order to reduce the radical dose of radiotherapy and/or to omit concurrent chemotherapy. This may decrease the acute and long-term toxicities of chemoradiation in patients with early cancers and/or low-risk disease such as HPV-positive HNSCC, as this will help to optimise treatment strategies in a more personalised manner.

## Figures and Tables

**Figure 1 cancers-14-04694-f001:**
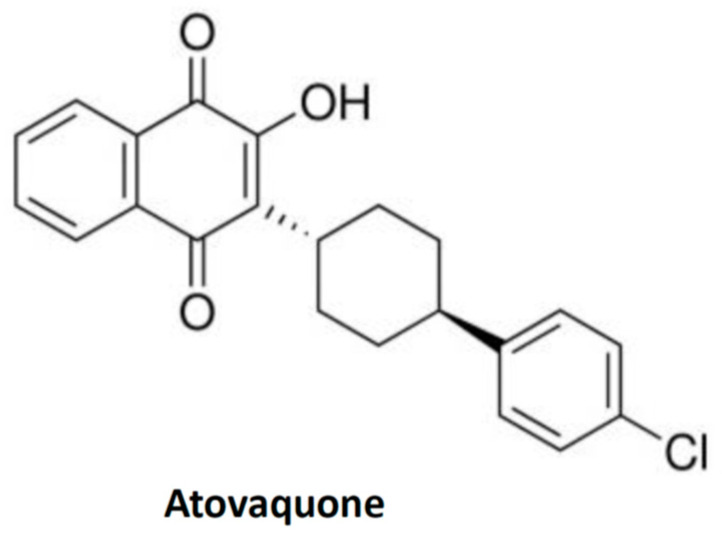
Structure of quinone-based compound, atovaquone. The structure is taken from the public repository PubChem structure for CAS 95233-18-4.

**Figure 2 cancers-14-04694-f002:**
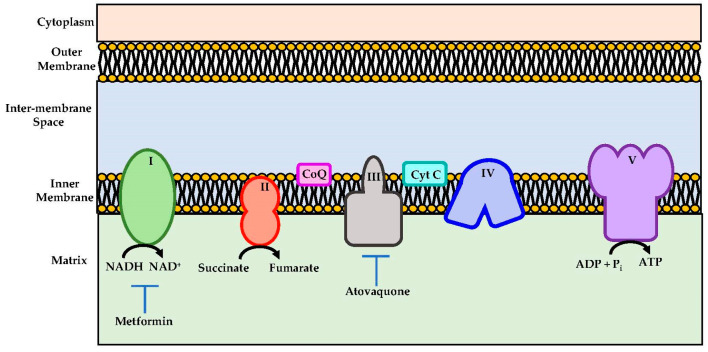
Schematic representations of mitochondrial electron transport chain. Metformin and atovaquone inhibits Complex I and complex III respectively and they are indicated as potential therapeutics importance.

**Table 1 cancers-14-04694-t001:** Comparison of different imidazole-based compound in clinical trials done in patients with HNSCC.

Compound	NCT Identifier (Ongoing Studies)	Clinical Trial	Study Design	Per Dose	Route of Administration	Other Treatment Modalities	Study Assessment	Outcome of the Study	Reference
Metronidazole	-	Oral cavity epidermal Carcinoma	Pilot study	6 g/m^2^	Orally	Interstitial irradiation	Radiosensitization	69% had complete local regression	[29]
Etanidazole	-	HNSCC;	Randomized Phase II	2 g/m^2^	Orally	Radiotherapy	Radiosensitization	No difference in LRC and OS	[30]
Etanidazole	-	Locally advanced HNSCC	Randomized phase III (RTOG 85-27)	2 g/m^2^	Orally	Radiotherapy	Radiosensitization	No difference in LRC and OS	[31]
Misonidazole	-	Locally advanced HNSCC	Randomized Phase III (EORTC 22811)	1 g/m^2^	Orally	Radiotherapy (3 times per day)	Radiosensitization	No difference in LRC and OS	[32]
Misonidazole	-	Larynx/pharynx carcinoma	Randomized Phase III (DAHANCA 2)	11 g/m^2^	Orally	Radiotherapy (split course)	Radiosensitization	Improved LRC in pharyx but not overall group	[33]
Misonidazole	-	Unresectable HNSCC	Randomized Phase 1/II (RTOG 79-04)	1.5 g/m^2^	Orally	Radiotherapy	Radiosensitization	No significant difference in LRC	[34]
Nimorazole	-	Larynx/Pharynx carcinoma	Randomized Phase III (DAHANCA 5-85)	1.2 g/m^2^	Orally	Radiotherapy	Radiosensitization	Significant improvement in LRC	[35]
Nimorazole	-	Unresectable HNSCC	Single arm Phase II	12 g/m^2^, 0.9 g/m^2^ 0.6 g/m^2^	Orally	Radiotherapy (CHART)	Radiosensitization	LRC better than historical control	[36]
Nimorazole	-	Unresected HNSCC	Prospective observation	1.2 g/m^2^ 1 g/m^2^	Orally	Radiotherapy (HART)	Radiosensitization	LRC similar to historical control of chemoRT	[37]
Nimorazole	NCT01950689	HNSCC	Randomized Phase III (NIMRAD)	1.2 g/m^2^	Orally	Radiotherapy	Radiosensitiz ation	Ongoing	-
Nimorazole	NCT01880359	HNSCC	Randomized Phase III	1.2 g/m^2^	Orally	Radiotherapy + cisplatin chemo	Safety, hypoxia radiosensitization	Ongoing	-

LRC: locoregional control; OS: overall survival; chemoRT: chemoradiation; CHART: continuous hyperfractionated accelerated radiotherapy; HART: hyperfractionated accelerated radiotherapy.

**Table 2 cancers-14-04694-t002:** Clinical trials of Atovaquone in other cancers.

Compound	NCT Identifier	Clinical Trial	Study	Doses of Atovaquone	Route of Administration	Other Treatment Modalities	Study Assessment	Outcome of the Study	Reference
Atovaquone	NCT02628080	Locally advanced NSCLC	Window of Opportunity study	750 mg/5mL	Orally	-	Hypoxia modifier	Reduced tumour hypoxic volume	[38]
Atovaquone	NCT04648033	NSCLC	Phase 1 (ARCADIAN)	750 mg/5mL	Orally	Radiotherapy+ concurrent chemo	Radiosensitization	Ongoing: to assess maximum tolerated dose (MTD)	-

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
