# Peer review of "The Therapeutic Potential of Imidazole or Quinone-Based Compounds as Radiosensitisers in Combination with Radiotherapy for the Treatment of Head and Neck Squamous Cell Carcinoma"

_cancers, 2022, doi:10.3390/cancers14194694_

Round 1

Reviewer 1 Report

This is an interesting review on the use of imidazole and quinones as radiosensitizers for the treatment of HNSCC and gives a nice overview of findings so far and the limitations.

I would suggest to carefully check again for missing words and punctuation marks, as I came across a lot of them during reading.

Furthermore, i would suggest to add some more citations between line 135-153 to strengthen the statements made in this part of the article.

Author Response

We thank the reviewer for the comments. We agree that there were missing words and punctuation marks and have gone through carefully to correct them. We have added more citations citations between line 135-153 to strengthen the statements made in this part of the article.

Reviewer 2 Report

Authors Azad and Kong have reviewed literature pertaining to the inclusion of quinone based compounds as radiosensitizers in HNSCC. they have included sufficient information for this to be accepted as is. 

Author Response

We thank the reviewer for the comments and for accepting the manuscript with no revision.

Reviewer 3 Report

This is an interesting review about the therapeutic potential of imidazole or quinone based compounds as radiosensitizers in combination with radiotherapy for the treatment of head and neck squamous cell carcinoma.

The paper is well written. However, some issues remain.

Since previous studies had not checked hypoxia status of the tumor following administration of metronidazole, the authors can not include such drugs among those that reduce hypoxia. Please better discuss this concept.

A table summarizing studies results should be added to help the readers.

Author Response

We would respectfully disagree with this reviewer. Our review is on the role of imidazole or quinone based compounds as radiosensitizers regardless of hypoxia reducing mechanisms or other mechanisms. Since metronidazole is an imidazole and was previously tested in clinical trials as radiosensitizer, we felt that it would be informative to include metronidazole. We have put in the text that “Based on the limited clinical studies, we cannot rule out metronidazole as a useful radiosensitizer and further detailed evaluation of metronidazole as a radiosensitizer with its machanism may be required in HNSCC, especially in combination with current intensity modulated radiotherapy (IMRT)”.

We thank the reviewer’s suggestion and have now included Table 1A and Table 1B summarizing studies’ results.

Reviewer 4 Report

we read with interest the review by Azad et al titled " The therapeutic potential of imidazole or quinone-based compounds as radiosensitizers in combination with radiotherapy for the treatment of head and neck squamous cell carcinoma" where the authors discuss the use of either imidazole or quinone based compounds as radiosensitizers or anti-cancer therapies focusing on the FDA approved anti-malarial drug atovaquone as an alternative to concurrent platinum chemotherapy in combination with radiotherapy. The work will discuss the treatment strategy of HPV negative HNSCC.

The work is of interest to researchers and clinicians there are some major comments that will enhance the work value.

First, the authors mention the clinical and preclinical studies involved in the imidazole or quinone-based compounds it would be essential to present these studies in comprehensive tables and show outcomes and their initial intended use even prior to being repurposing to being adjuvant to cancer therapy.

The authors should dedicate a section on the ischemic pathways involved that the proposed imidazole or quinone-based compounds would target and highlight why it is a major pathological pathway in cancer relapsing

Table 1 presented has limited value as it lacks references and it has no outcomes and treatment modalities, treatment dosages and route of administration, and most importantly the outcomes with comments on these studies.

There is a missing section on limitations and the lag in translating these drugs into clinics where the authors should discuss the side effects of imidazole or quinone-based compounds.

Please insert a mechanistic schematic that illustrates how these compounds target the HNSCC; Fig 1 has low importance

Minor Comments:

English Editing is needed, the writing has some run-on sentences such as: “have been shown to hypoxia and make radiotherapy more effective”

please use : quinone-based

Author Response

We thank the reviewer for many helpful suggestions. We have now modified Table 1 to include treatment modalities, treatment dosages and route of administration, and the outcomes with comments on the clinical studies. We have described preclinical studies in the text as it would be difficult to put it in a table.

Although the nitroimidazole group of drugs are acting as oxygen mimetic, the detailed mechanism on how it causes reduction in hypoxia is not clearly known yet. We have stated this in the text.

Although this reviewer feels that the structure of atovaquone depicted in Figure 1 has low importance, we still feel that it is beneficial for interested reader. Atovaquone is thought to act as a potent and selective OXPHOS (oxidative phosphorylation) inhibitor, by targeting mitochondrial complex III. Atovaquone is structurally similar to the inner mitochondrial protein coenzyme Q (also known as ubiquinone), which is an integral component of electron transport chain in oxidative phosphorylation. Coenzyme Q accepts electrons from dehydrogenase enzymes and passes them to cytochrome bc1 (complex III) that requires binding of coenzyme Q-complex III at the Qo cytochrome domain; atovaquone inhibited this step of electron transport chain. The consequence of this inhibition is the collapse of the mitochondrial membrane potential and thus rendering apoptosis. We have depicted this mechanism in the newly included Figure 2. It was also shown that atovaquone could alleviate tumour hypoxia at a pharmacologically achievable concentrations by reducing oxygen consumption rate (OCR). By slowing down the use of oxygen, atovaquone reversed low oxygen levels in tumours. In addition to atovaquone, metformin, though not a quinone based compound, acts as an OXPHOS inhibitor through complex I inhibition and we have included in Figure 2 alongside with atovaquone to show the differences between the two drugs. Metformin was also shown to alleviate hypoxia in vivo through reducing OCR, which we stated in the text.

Hypoxia is well documented as one of the main resistance mechanisms to radiotherapy and chemotherapy resulting in treatment failure and cancer recurrence. We have mentioned in the review on possible targets of atovaquone ranging from oncogenic signalling and cell cycle. However, its mechanism in targeting hypoxia is likely due to reduction in oxygen consumption through inhibition of OXPHOS as described above.

In our review, we have discussed the limitations and side effects of the different imidazole and quinones as radiosensitizers. In addition, we compared them in our review and have focused on atovaquone as the promising and efficacious radiosensitizing agent with no significant toxicities and good safety profiles demonstrated in clinical trials thus far.